# Seroprevalence and Risk Factors Associated with *Leishmania* Infection in Dogs from Portugal

**DOI:** 10.3390/microorganisms10112262

**Published:** 2022-11-15

**Authors:** Maria Almeida, Carla Maia, José M. Cristóvão, Cátia Morgado, Inês Barbosa, Ruben Foj Ibars, Lenea Campino, Luzia Gonçalves, Sofia Cortes

**Affiliations:** 1Instituto de Higiene e Medicina Tropical (IHMT), Universidade Nova de Lisboa (UNL), Rua da Junqueira, 100, 1349-008 Lisboa, Portugal; 2Global Health and Tropical Medicine (GHTM), Instituto de Higiene e Medicina Tropical (IHMT), Universidade Nova de Lisboa (UNL), Rua da Junqueira, 100, 1349-008 Lisboa, Portugal; 3LETI Pharma S.L.U., 08038 Barcelona, Spain; 4MSD Animal Health Lda, 2770-192 Paço de Arcos, Portugal; 5Centro de Estatística e Aplicações da Universidade de Lisboa (CEAUL), Faculdade de Ciências, Universidade de Lisboa, 1749-016 Lisboa, Portugal; 6z-Stat4life, Cowork do Palácio Baldaya, 1549-011 Lisboa, Portugal

**Keywords:** leishmaniasis, dogs, *Leishmania*, seroprevalence, southwestern Europe, Portugal, direct agglutination test, risk factors

## Abstract

Canine leishmaniosis (CanL) caused by *Leishmania infantum* is an important zoonosis in southwestern European countries where this disease is endemic, and dogs, as domestic animals in close contact with humans, are the reservoir hosts for the parasite. In Portugal, CanL is of relevant veterinary concern. The previous national study revealed an overall seroprevalence of 6.3%. Since then, new prophylactic measures, such as vaccines, have been introduced in Europe. The aim of this study was to update seroprevalence for *Leishmania* infection and reassess risk factors in Portugal. A cross-sectional study was conducted from January–March 2021 with 1860 client-owned dogs from continental Portugal. A questionnaire and whole blood samples on filter paper were collected and a direct agglutination test was used to calculate anti-*Leishmania* antibody titres. True seroprevalence was 12.5% (95% CI 10.3–13.2%). Potential risk factors associated with *L. infantum* infection in dogs were age ≥ 2 years (aOR = 1.68, 95% CI 1.1–2.6) and residing in the interior regions of the country (aOR = 1.92, 95% CI 1.3–2.9) and non-use of repellents (aOR = 1.75, 95% CI 1.2–2.5). The key to controlling CanL and its impact on Public Health in endemic areas lies in continuous implementation of prophylactic measures, through the correct use of repellents/insecticides and vaccines and early detection and monitoring of infected dogs.

## 1. Introduction

Canine leishmaniosis (CanL) is a zoonosis endemic in many regions caused by the protozoan parasite *Leishmania infantum*. Domestic dogs are the parasite’s principal hosts and reservoirs for human visceral leishmaniasis, a fatal illness if left untreated. The parasite is transmitted by female phlebotomine sand flies of the genera *Phlebotomus* in the Old World and *Lutzomyia* in the New World [1]. The close contact of dogs with humans makes this zoonosis a public health concern. CanL affects millions of dogs and is largely endemic in two main regions: South America and the Mediterranean basin, spanning approximately 50 countries [2]. In the Mediterranean region, CanL has a much higher incidence than human leishmaniasis [3]. Southern European countries (France, Greece, Italy, Spain and Portugal) have been endemic for CanL for a long time [4,5,6]. A recent study addressing the geographical distribution of CanL incidence in Spain and France confirmed the northward spread of CanL in these countries [7]. Moreover, in Europe, dogs from non-endemic countries travelling with their owners to southern Europe or dogs adopted from these regions pose a risk for the emergence of leishmaniosis in non-endemic temperate regions of northern Europe [8,9]. The leishmaniosis scenario is changing in part due to climate changes in the Mediterranean region and in Europe, which consequently influences phlebotomine distribution [10]. CanL prevalence varies greatly over time and between adjacent locations, as they are influenced by bioclimatic factors affecting vector abundance. In addition, infected asymptomatic dogs, which can reach up to 50%, are infectious to sand flies, enabling the parasite’s life cycle to continue and therefore allowing other animals, as well as humans, to be infected [11].

In Portugal, human visceral leishmaniasis is hypoendemic; as a disease of compulsory notification, the last data reports an incidence rate of 0.1 cases per 100,000 inhabitants in 2017 [12]. On the other hand, CanL is considered a veterinary concern and is present throughout the country with a heterogenous distribution [13]. Recently, in the northeastern region of Portugal, a seroprevalence for *Leishmania* infection of 21.3% was found [14]. But much higher values were found in the Interior Centre of Portugal, Castelo Branco, with seroprevalences reaching 56.0% in some municipalities [14]. High prevalence was also identified in the south of Portugal, with 60.4% dogs positive for *L. infantum* using molecular methods [15]. The last national epidemiological survey, performed more than a decade ago on 3974 dogs, identified a seroprevalence of 6.31% for *Leishmania* infection, with Interior Districts reaching seroprevalences of up to 17.4% [13]. Concerning the Portuguese archipelagos, Madeira and the Azores, no autochthonous cases are known, and detected cases are imported from continental Portugal [16].

Clinical indicators of CanL, a systemic chronic disease, include generalized lymphadenomegaly, skin lesions, weight loss, ocular signs, epistaxis, lameness and onychogryphosis [1,2]. The diagnosis of CanL is complex and multiple parameters need to be considered. Serological procedures are among the most frequent diagnostic techniques used to detect *Leishmania* infection in both ill and subclinical infected dogs in endemic regions. The indirect fluorescence antibody test (IFAT), ELISA and rapid tests are frequently used for diagnosis [17,18]. The direct agglutination test (DAT) is also used mainly for epidemiological surveys, with sensitivities of 93–100% and specificities of 95–99% [19,20,21]. Control measures for *Leishmania* infection rely mainly on the application of insecticides/repellents against sand flies, with pyrethroids, permethrin, or deltamethrin as the formulations. In the last decade, two vaccines have been implemented in the European Union (EU): CaniLeish^®^ (Virbac, Carros, France) and LetiFend^®^ (LETI Pharma, Barcelona, Spain) [22]. Several studies have demonstrated that CaniLeish^®^ presents limitations for serological diagnosis and does not differentiate between infected and vaccinated animals, as anti-*Leishmania* antibodies may be detected in both naturally infected and vaccinated dogs [22,23]. In 2022, CaniLeish^®^ was withdraw from the EU market. LetiFend^®^ is considered a vaccine that enables differentiating infected from vaccinated animals (DIVA) and does not interfere with antibodies measured by different serological approaches [22]. Nevertheless, the implementation of CanL vaccines in the last decade in Europe has brought concerns that are still unclear—namely, whether vaccinated but infected dogs are able to reduce their infectivity or continue to serve as a reservoir for *Leishmania* parasites and phlebotomine feeding, as do unvaccinated animals [24].

The aim of the present study was to update the status of *Leishmania* infection in dogs from continental Portugal through a cross-sectional serological survey performed in 2021 and to identify risk factors and compare the data with a similar survey conducted in 2009, keeping in mind the introduction of canine vaccines in the last decade.

## 2. Materials and Methods

### 2.1. Study Area

Portugal is located in Southwest Europe, bordering the Atlantic Ocean. The littoral zone is characterized by mild summers, whilst the interior region is characterized by hot and dry summers. Continental Portugal is divided into 18 districts spread over five geographical regions (nomenclature of territorial units for statistics, NUTS2 subdivisions): North, Centre, Lisbon and Tagus Valley, Alentejo and Algarve. Viana do Castelo, Braga, Porto, Aveiro, Coimbra, Leiria, Lisboa, Setúbal and Faro are considered littoral districts, whereas Vila Real, Bragança, Viseu, Guarda, Castelo Branco, Santarém, Portalegre, Beja and Évora are interior districts.

### 2.2. Survey Strategy and Sampling

A cross-sectional national survey was performed in continental Portugal on client-owned dogs. In the beginning of 2020, veterinary clinics (“Centros de Atendimento Medico-Veterinário”) were invited to participate in a national canine survey, an online survey provided by “Ordem dos Médicos Veterinários”. Initially, 135 clinics agreed to participate, but due to the beginning of the COVID-19 pandemic and a nationwide lockdown, the study was postponed to January 2021. When contacted again, a total of 98 veterinary clinics agreed to participate in this study (Appendix A).

A stratified proportional sampling by district was implemented, using data from the Portuguese Information System of Companion Animals (“Sistema de Informação de Animais de Companhia”, SIAC). Epitools^©^ Epidemiological Calculators [25,26] was used to estimate the sample sizes based on a 95% confidence interval (CI) for prevalence with a precision of 3%, considering the use of the DAT test, with a specificity and sensibility of 100% and 93%, respectively [20]. Moreover, estimates on the true prevalence for each region were based on the previous Portuguese national survey [13]: North: 3.8%, Centre: 7.8%, Lisbon and Tagus Valley: 6%, Alentejo: 10% and Algarve: 4.7%. The estimated sample sizes per region were as follows: North 177, Centre 340, Lisbon and Tagus Valley 261, Alentejo 417 and Algarve 219, with a total of 1414 canine samples (Appendix A).

Although a minimum sample size of 1414 was established for this study, three times more sample kits were supplied to the veterinary centers in order to reduce fieldwork losses. An explanatory letter and a poster were sent to each clinic, along with a sampling collection kit consisting of the following: (i) numbered filter paper in a small plastic bag for sample collection and (ii) informed consent and (iii) a questionnaire (Appendix A). The epidemiological variables were as follows: geographic location (region, district, county), age, sex, breed, dog’s housing, use of protective measures and presence of clinical signs compatible with CanL. The study was advertised to the dog’s owners through posters and social media posts done by the veterinary clinics. Sample collection randomization was requested of the participating clinics, namely the following: (i) the first animals that come at the clinic on the day of the initiative and whose owner give consent for sampling and (ii) a fixed number of animals should be sampled each day. Exclusion criteria were set as follows: (i) dogs younger than 6 months old and (ii) dogs vaccinated with Canileish^®^ for less than 6 months before the sampling. Samples were collected from January to March 2021.

### 2.3. Sample Collection and Serological Test

A coin-sized whole blood sample (200–400 μL) was collected onto filter paper (Whatman^®^ Grade 3, Sigma, Darmstadt, Germany) through venipuncture of the dog. Samples were dried at room temperature and sealed in their respective plastic bags, sent back to Instituto de Higiene e Medicina Tropical-Leishmaniasis laboratory (IHMT) and kept at room temperature.

DAT was used for the detection of anti-*Leishmania* antibodies [19]. A 5.5 mm diameter disk was collected with a paper punch from the filter paper blood spot and placed on the first column of a V-shape 96-microwell plate (BRAND^®^, Wertheim, Germany), then eluted overnight in 120 μL saline solution (NaCl 0.9%, *w*/*v*, Sigma, Germany) at room temperature. The dilution solution was prepared with NaCl 0.9% and β-mercaptoethanol 0.2 M (Sigma) to perform twofold serial dilutions ranging initially from 1:50 to 1:800 in a final volume of 50 μL (in accordance with manufacturer instructions, the blood eluted from the 5.5 mm disk diameter is equivalent to 5μL, which enables an initial blood dilution of 1:25). The freeze-dried antigen (KIT Biomedical Research, Amsterdam, The Netherlands) was eluted in NaCl 0.9% and 50 μL were placed in each well according to the manufacturer’s instructions. All plates included a positive control (sample from a dog positive by culture and PCR positive) and a negative control (sample from a healthy dog with no positive tests or evidence of disease). Test plates were left at room temperature for 18 h and the results were observed for the presence (positive) or absence (negative) of aglutination.

A cutoff titer (threshold of positivity) of 400 was considered to maximize specificity and sensitivity [27]. Samples with a titer equal to or higher than 400 were repeated with serial twofold dilutions ranging from 1:50 to 1:25 600 to set the definitive titer.

### 2.4. Statistical Analysis

Questionnaire data and statistical tests were analyzed using IBM^®^ SPSS^®^ Statistics Version 27. Categorical variables collected on the questionnaire regarding the dog’s features were sex (male; female), age (five classes: 0.5–2, 3–5, 6–8, 9–11 and 12–17 years), fur size (short; medium; long), and breed (non-autochthonous pure breed; autochthonous pure breed; crossbreed; mongrel dogs). Information regarding district and geographical area (littoral; interior), dog’s housing (exclusively outdoors; mostly outdoors; equally indoors and outdoors; mostly indoors; exclusively outdoors), use of repellents/insecticides (effective; non-effective; used but unknown; nonuse), vaccination status and vaccine brand against CanL (vaccinated with LetiFend^®^; vaccinated with CaniLeish^®^; vaccinated (unknown vaccine); nonvaccinated) was also collected on the questionnaires. Effective repellents/insecticides were those with pyrethroids, permethrin, or deltamethrin in their composition. Information on the use of any kind of medication and clinical signs compatible with CanL (yes; no) was also collected. Descriptive statistics were applied to summarize each mentioned categorical variable, using absolute and relative frequencies.

A chi-square test was used to determine associations between categorical variables associated with outcome and other variables with relevant epidemiological interest. This test is also used to compare prevalence among regions or other independent groups. Fisher’s exact test was used in case of failure of the assumptions of the chi-square test. A multivariate analysis was conducted through multiple binary logistic regression models, analyzing variables with statistical meaning in the univariate analysis and some potential confounding variables. For those variables that remained significant, crude odds ratio (OR) were updated to adjusted odds ratio (aOR) with 95% confidence intervals (95% CI). The Hosmer–Lemeshow test was used for assessing goodness of fit in each multiple logistic regression model [28]. Reference categories for each variable were used in the analysis to determine risk factors for positivity. The reference categories used for statistically significant variables were as follows: two or older (recategorized) for the variable “Age”, interior for “Interior/Littoral”, Alentejo for “Geographical Area” and non-use for “Repellents/ insecticides”.

For proportions, the Wilson method was used to obtain 95% confidence intervals. True prevalence was calculated based on the following formula: true prevalence (TP) = (test prevalence—1 + specificity)/(sensitivity—1 + specificity) considering 95% sensitivity and 100% specificity for the test considered (DAT) [20]. The corresponding 95% confidence intervals were obtained using Epitools^©^ Epidemiological Calculators (https://epitools.ausvet.com.au/, accessed on 5 June 2022) [25,29]. The true seroprevalence values for *Leishmania* infection in dogs according to Portuguese districts and NUTS2 were presented in a heat map with categorical intervals drawn in mapinseconds.com (accessed on 16 October 2022).

### 2.5. Ethical Clearance

This study was approved by the Animal Welfare and Ethics Committee of IHMT (“ORBEA”) and the National Official Authorities for animal experimentation (“Direção Geral de Alimentação e Veterinária”, Portugal) with reference 0421/000/000/2021. All dog owners were informed about the study protocol and signed an informed consent form allowing for sample and data collection.

## 3. Results

### 3.1. Seroprevalence According to Geographic Regions

A total of 1877 canine blood samples on filter paper were collected from the 18 continental Portuguese districts. From these, 17 samples were excluded due to the pre-determined exclusion factors for the study or because they were misidentified, with 1860 validated blood samples and questionnaires. For some variables, missing data was observed on the questionnaires.

A total of 98 veterinary clinics were enrolled in the survey. The number of clinics per district varied from two (Guarda, Portalegre) to 12 (Lisboa) (Table 1). Based on the previous estimated number of samples per region, Alentejo, with 247 samples, and Algarve, with 135 samples, were underrepresented, as the initial estimated numbers of required canine samples were 417 and 219, respectively. A total of 57.8% (*n* = 1076) of the dogs were living in littoral districts, whereas 42.2% (*n* = 784) were from the interior of the country. In terms of district distribution, the highest number of dog samples came from Lisboa (*n* = 210), and the lowest number of collected samples came from Guarda (*n* = 40) (Table 1). The highest percentage of vaccinated dogs was found in Faro (32.1%) and Setúbal (27.4%), and the lowest in Viana do Castelo (0.0%) and Vila Real (2.9%) (Appendix A).

Overall, 217 dogs were positive for the presence of anti-*Leishmania* antibodies (11.7%) considering the DAT cut-off titre equal or higher to 400 (Table 1). A total of 61.8% (134/217) of positive samples presented a titre equal to 25 600 (Appendix A).

Considering the whole sample set (*n* = 1860), the overall true seroprevalence for *Leishmania* infection in Portugal was 12.5% (95% CI 10.3–13.2). True seroprevalence per region (NUTS2) varied from 9.6% (95% CI 7.2–12.8) in the North to 17.2% (95% CI 11.8–24.7) in Algarve. The proportion of positive dogs from the interior was significantly higher than those living in regions closer to the littoral (*p* < 0.001). True seroprevalence varied from 0% (95% CI 0.0–7.5) in Viana do Castelo, in the North of Portugal, to 30.5% (95% CI 19.9–43.8) in Portalegre, followed by Castelo Branco and Guarda with 29.9% (95% CI 20.1–42.0) and 19.3% (95% CI 9.6–35.1), respectively. Figure 1 shows the 18 Portuguese districts and true seroprevalence distribution.

As CaniLeish^®^ vaccine elicits vaccinal antibodies and just recently this vaccine was withdrawn from the EU market, true seroprevalence was re-evaluated not considering positive dogs with this vaccine. The observed overall true seroprevalence was 11% (95% CI 9.6–12.5), slightly lower when considering the overall sample set (Appendix A). Nevertheless, districts with higher and lower seroprevalences maintained their proportions.

### 3.2. Seroprevalence According to Dogs’ Characteristics

The percentage of female (*n* = 944, 51.5%) and male (*n* = 890, 48.5%) dogs was similar (Table 2) and equally tested positive, with no significant differences between sample sets. The dogs’ age varied from six months (0.5 years old) to 17 years old. More than half of the dogs were aged between 0.5 and 5 years old, with a mean age of 5.39 ± 3.76. The age group 12–17 had the highest proportion of positives with 15.4% (95% CI 0.0–31.2), and the 0.5–2 age group had the lowest proportion, with significant statistical differences between all groups (*p =* 0.005).

Regarding breed, most of the dogs were from non-autochthonous pure breeds (*n* = 817, 43.9%) followed by mongrels (*n* = 774, 41.6%). Dogs from autochthonous pure breeds presented a higher rate of positivity with 14.4% (15/104) but with no significant statistical differences in comparison with non-autochthonous or mongrel dogs. The most frequent non-autochthonous breeds were Labrador Retriever, German Shepherd and French Bulldog, whereas Podengo, Rafeiro Alentejano and Serra da Estrela were the most frequent autochthonous breeds (Appendix A). Although more than half of the dogs had short fur (*n* = 1109, 60.2%), the dogs’ positivity concerning fur size was similar between the three variables. The majority of dogs were kept mostly indoors (*n* = 637, 34.2%), followed by dogs living exclusively outdoors (*n* = 396, 21.3%). No significant differences were found regarding the dogs’ positivity and the dogs’ housing.

Concerning use of preventive measures, 69.6% (1269/1825) of the dogs used repellents. Effective repellency against sand flies or in reducing *L. infantum* infection was reported in 40.1% (*n* = 732) of the cases, sometimes combined with other non-effective insecticides, and 30.5% (*n* = 556) did not use any repellent/insecticide on their animals (Table 2). A higher percentage of positive dogs was observed in the group not using any repellent/insecticide (13.1%; 95% CI 5.3–20.8), although similar to the ones using effective products (12.3%; 95% CI 5.5–19.1). The most-used repellents were imidacloprid/flumethrin (Seresto^®^, Elanco, Greenfield, IN, USA), imidacloprid/permethrin (Advantix^®^, Bayer, Leverkusen, Germany) and deltametrin (Scalibor^®^, Merck Animal Health, Madison, NJ, USA). Fluralaner (Bravecto^®^, Merck Animal Health), a systemic insecticide and acaricide that is administered orally, was also frequently reported. The frequency of application of the various repellents/insecticides ranged from monthly (*n* = 350), quarterly (*n* = 385), every four to six months (*n* = 37), every eight months (*n* = 342) to annually (*n* = 342). In many cases, periodicity did not correspond to the recommendations of the brands. In addition, 14.9% of dogs (271/1824) were vaccinated, and the most prescribed vaccine was LetiFend^®^ (*n* = 142, 7,8%). Seropositivity of dogs vaccinated with CaniLeish^®^ (30/63, 47.6%; 95% CI 29.7–65.5) was much higher than with those vaccinated with LetiFend^®^ (26/142, 18.3%; 95% CI 3.4–33.2) (Table 2).

The presence of clinical signs compatible with CanL was reported in 6.2% (112/1804) of dogs and the seropositivity within this group was 37.5% (42/112), but 2.3% (42/1804) considering all sample set. The most reported clinical signs on the questionnaires were the following: loss of appetite/body weight, ulcers, wounds and other unspecified skin lesions, alopecia, onychogryphosis, and hepatic alterations (Appendix A). Analysing vaccination status and the presence of clinical signs compatible with CanL within the seropositive dogs, it was observed that the group of seropositive non-vaccinated dogs presented significantly more animals with clinical signs (26%) than the seropositive vaccinated dogs, with only 7% (*p =* 0.002) (Figure 2).

### 3.3. Associated Risk Factors for Leishmania Infection in Dogs

Analyzing variables excluding vaccinated dogs, the risk factors identified were “Older than 2 years”, “Residing in the Interior” “Residing in Alentejo” and “Nonuse of repellents/insecticides”, with significant statistical differences (*p* ≤ 0.05) (Table 3). A multiple logistic regression model again identified age, region of the country and nonuse of repellents/insecticides as predictors for *L. infantum* infection in exposed dogs. The chance of infection increased in dogs older than two years (aOR = 1.68, 95% CI 1.1–2.6, *p* = 0.02), in dogs residing in the interior of the country (aOR = 1.92, 95% CI 1.2.9, *p* = 0.002) and in dogs not using repellents/insecticides (aOR = 1.75, 95% CI 1.2.5, *p* = 0.003).

## 4. Discussion

Although parasites of the genus *Leishmania* sp. can infect a variety of vertebrate animals, dogs are the main reservoir for zoonotic human leishmaniasis caused by *L. infantum* in the Mediterranean basin. Infection in dogs registers a much higher prevalence with infection rates up to 60% in exposed populations compared to human infection, which is hypoendemic in Europe [3,30].

The goal of this study was to update the seroepidemiological status of *Leishmania* infection in dogs from Portugal, as well as to determine the current risk factors following the adoption of new preventative measures in Europe, such as new insecticide formulations and vaccines. Different variables were analyzed, such as age, sex, breed, fur size, dog’s housing, location (geographical region and district), use of repellents/insecticides and vaccination status.

In Portugal, canine epidemiological studies in different regions have been performed over the years [14,15,31]. The last national seroepidemiological survey was performed more than one decade ago. In 2012, overall true seroprevalence was 6.5% using DAT as the serological test, with interior districts presenting higher seroprevalences [13]. In the present national survey, the overall true seroprevalence found was 12.5% (95% CI 10.3–13.2) (11.0%, 95% CI 9.6–12.5, not considering CaniLeish^®^-vaccinated dogs), and, again, the districts with the highest seroprevalences were Portalegre (30.5%, 95% CI 19.9–43.8), Castelo Branco (29.9%, 95% CI 20.1–42.0) and Guarda (19.3%, 95% CI 9.6–35.1). Seroprevalences up to 56.0% were found in a recent study conducted in several municipalities of the Castelo Branco District, reaffirming once again that this is a highly endemic region for CanL [14]. Despite the fact that Portugal is a small country, we cannot rule out the possibility that dogs may travel with their owners during the summer months or hunting season, which coincide with the sand fly transmission season, to other districts and become infected away from their residence. Because the information gathered through questionnaires in this study only relates to dogs who live close to each veterinary clinic, it was not possible to assess the transregional influence on seroprevalence.

In the present study, interior districts exhibit a higher seroprevalence for *Leishmania* infection than littoral districts, posing an increased risk for dogs being infected (aOR = 1.92, 95% CI 1.3–2.9, *p* = 0.002), as observed in the past. Although a geospatial analysis and land-types analysis were not performed, dogs from interior districts, such as Portalegre, Castelo Branco and Guarda are prone to live in a more rural environment, such as peridomestic biotopes, particularly close to sheep pens, cattle stables, and henhouses, with possible accumulation of organic matter, which may provide favourable conditions for sand flies’ resting and breeding sites [32]. As there is a lack of phlebotomine studies in districts where higher seroprevalences are observed, it is difficult to assess the influence of the phlebotomine season and densities on the present results. In the future, it would be important to assess their presence and identify the species responsible for the circulation of the parasites (either *Phlebotomus ariasi* or *P. perniciosus*). According to a study on the seasonal dynamics of sand fly species in the Mediterranean region including Portugal, the distribution and abundance of *P. perniciosus* varied over the course of three years (2011–2013), with seasons that last from April till October, and in some cases up to November [33].

In Spain, a study performed between 2011–2016 on 1739 dogs from 25 Spanish provinces using IFAT as the serological technique identified a seroprevalence for *L. infantum* infection of 10.1%, and the most abundant sand fly was *P. perniciosus* [5]. Moreover, Le Rutte et al. [7] found high CanL incidence rates in southeastern Spanish districts with a growing trend over time. Some of these regions are close to the Portuguese districts with higher seroprevalences for *L. infantum* infection. Residing in interior regions at higher altitudes, with warm average temperatures in the Mediterranean Basin, has been identified as a risk factor for CanL [5,7].

Faro District in the southern region of Algarve presented a CanL seroprevalence of 17.5% (95% CI 11.8–25.2), similar to the 18.2% found previously in this region also using DAT [15]. A study on the same region presented substantial differences in dogs’ positivity when molecular methods were applied for the detection of *Leishmania* DNA by PCR, with positivity reaching 69% in dogs without clinical signs compatible with CanL [34]. These differences can be partially explained by the methodology used but also by the population sampled, as these two studies also included stray dogs from shelters. With molecular techniques, sensitivity can be much higher and allows the detection of the parasites even in seronegative dogs [35]. A limitation of the present study was the sampled dogs’ population, as the samples were obtained from veterinary clinics, in client-owned dogs with access to veterinarian care. As the sample is not quite representative of Portugal’s dog population, due to the lack of stray and sheltered dogs, probably the scenario is worse.

No differences were found in seropositivity among mongrels and pure or cross breeds, as also observed by others using ELISA and DAT as serological approaches [36,37]. On the other hand, studies have shown that the breed may be associated with resistance or susceptibility to *L. infantum* infection. Solano-Gallego et al. [38] observed a significant cellular immune response against infection in Ibizian hounds with a resistant pattern. Mongrel dogs or crossbreeds were also considered less susceptible to infection in comparison with pure exotic breeds [13,39]. Contrary to the previous national seroepidemiological study, fur size seemed not to affect dogs’ positivity, meaning that the vector is biting dogs with long, medium and short fur. The vector is known to have a preference for muzzles, due to the CO_2_ presence, and ears, which in most cases are not protected by fur [2].

In this study, being older than two years was considered a potential risk factor for seropositivity in dogs (aOR = 1.68, 95% CI 1.1–2.6, *p* = 0.02). Different studies show that age can be a risk factor for *Leishmania* infection [40], with older dogs more prone to infection [39,41,42]. Others found higher positivity rates in younger dogs (<1 year-old) [5]. On the other hand, Symeonidou et al. [43] and Coura-Vital et al. [44] found no association with presence of anti-*Leishmania* antibodies and age, using, respectively, a rapid test (i.e., SpeedLeish^®^) and ELISA. Although some studies have shown that age distribution and seropositivity follow a bimodal distribution [37,45], we have not observed this trend. A possible explanation could be the age class stratification.

Preventive measures for CanL involve, among others, keeping dogs in the house from dusk to dawn, the use of effective repellents/insecticides and vaccination. Less than half of dogs were using effective repellents against the sand fly vector or in reducing *L. infantum* infection, in many cases combined with other ineffective products (40.1% of dogs). Curiously, the percentage of seropositive dogs not using any insecticide (13.1%; 95% CI 5.3–20.8), and the percentage of seropositive dogs using effective ones (12.3%; 95% CI 5.5–19.1) were similar. It is possible that these products were not being correctly applied. The same trend was observed by others both in Portugal and in Spain, where the use of specific prophylaxis against CanL failed to show a protective effect [34,37]. This may be explained with the lack of compliance by owners when applying the preventive products on their dogs. In a recent study based on an online questionnaire sent to veterinary clinics in Europe analyzing strategies for preventing *L. infantum* infection or clinical disease in dogs, it was shown that repellents (86.2%), vaccination (39.8%) and Leishguard^®^ (15.3%) were the most frequently employed [46].

In the present study, it was observed that in many dogs the application interval of insecticides was not in accordance with brand recommendations, varying from two months to a year for one brand (data not shown). Thus, the improper use of repellents, lack of compliance with the instructions of the brands, and not considering the transmission season may explain the increase on seroprevalence when compared with ten years ago and why the nonuse of repellents/insecticides was identified as a risk factor (aOR = 1.75, 95% CI 1.2–2.5, *p =* 0.003). An online survey performed among veterinarians from Spain and France revealed that these professionals in endemic regions recommend preventive measures to most of their clients [47].

In recent years, two vaccines have been commercialised in Europe, CaniLeish^®^ and LetiFend^®^. In this study it was observed that only 14.9% of dogs were vaccinated (271/1824). This fact is likely related to the costs of the vaccines. LetiFend^®^ was the most frequently applied (7.8%), probably due to its single-dose primary vaccination schedule and to its very good safety profile [48]. In contrast, CaniLeish^®^ is prescribed in a three-dose primary vaccination [49]. Moreover, it is known by the veterinary community that CaniLeish^®^ may cause adverse effects on dogs [23]. In 2022, CaniLeish^®^ was withdrawn from the market (https://www.leishvet.org/fact-sheet/vaccines/, accessed on 2 September 2022).

Like IFAT, DAT is based on whole *Leishmania* antigen prepared from freeze-dried *L. donovani* promastigotes [19] and detects total antibodies. In this study, to reduce the cross-reaction of vaccinal antibodies, dogs vaccinated with Canileish^®^ for less than six months were excluded, but it is likely that vaccinal antibodies induced by CaniLeish^®^ were being detected after six months. Therefore, this serological test may result in false positives for CaniLeish^®^-vaccinated dogs, as previously observed in other serological studies. Velez et al. [24] observed cross-reactivity by in-house ELISA in CaniLeish^®^-vaccinated dogs one month after the end of the vaccination protocol. Another recent study aiming to evaluate antibody kinetics in CaniLeish^®^-vaccinated dogs using IFAT and SpeedLeish K^®^ showed that in 3.2% of dogs, vaccinal antibodies were still detected after one year [50]. LetiFend^®,^ is a DIVA vaccine, being expected to only elicit antibodies against protein Q, which is a recombinant protein made of different proteins’ fragments derived from *L. infantum* [24]. Thus, it does not interfere with *L. infantum* antibodies measured by the most widely used quantitative serological diagnosis tests, such as IFAT and ELISA [22,24]. Nevertheless, further studies should be performed in a long-term course using DAT in parallel with other methods to assess if there is any interference associated with LetiFend^®^. The high seropositivity rates in the vaccinated dogs may be a confounding factor, on one side because of the cross-reactivity in the case of CaniLeish^®^, but also due to a lack of preventive measures or low owner compliance with repellent application rather than to the lack of vaccine efficacy. Also, in districts with high prevalence, veterinarians are more aware of the risks of infection and recommend more vaccination to prevent development of CanL, which may explain the higher positivity rates in LetiFend^®^-vaccinated dogs because, although vaccinated, these animals are more exposed to the parasite. The primary role of vaccination is to decrease the chance of developing clinical disease rather than to provide protection against parasite exposure [22].

Even given the previous limitations concerning vaccinated dogs, DAT was the serological technique selected for this study, as it is simple, inexpensive, can be applied to many samples, does not require any equipment and has a track record of clinical accuracy for both canine and human serological diagnosis [19,36,51]. Moreover, the choice of this serological approach was also to compare the trend of seroprevalence in Portugal after more than 10 years since the last national seroepidemiological canine survey [13] where the same approach was used. PCR is a more sensitive and specific diagnostic method than serological approaches for the detection of CanL, but more invasive procedures, such as lymph node or bone marrow aspiration, are required for a precise diagnosis [22]. Although beyond the purview of this epidemiological study, it was strongly recommended that seropositive dogs exhibiting clinical signs be followed in accordance with Leishvet recommendations [35] to confirm infection and carry out further treatment.

In the whole sample set, 19.3% of seropositive dogs (42 out of 217) presented clinical signs. The most common clinical signs were skin lesions, weight loss, onychogryphosis, and alopecia, as observed in other studies [4,42,52]. Despite the fact that some vaccinated dogs were seropositive, only 7% (5/72) had CanL-consistent clinical signs, in contrast to seropositive nonvaccinated dogs, which showed a substantially higher proportion (26%, 37/142, *p* = 0.002). This finding supports the vaccine’s major role in reducing the risk of developing a clinical disease after exposure to *L. infantum* rather than reducing sand fly contact and further infection.

This study highlights the need for monitoring *Leishmania* infection and CanL in endemic countries and the need to promote more initiatives on awareness for this zoonosis and its control measures. A study from 2018 investigated the implementation of these guidelines among veterinarians in Spain and France and found that although 60% of the 459 veterinarians were aware of the current increase of *L. infantum* in Europe, 70% were not familiar with any guidelines for its control [47].

## 5. Conclusions

Overall, the present study shows that dogs are highly exposed to the vector and consequently to *Leishmania* infection, and Portugal continues to be an endemic country for CanL with increased seroprevalences in the entire territory. These findings reaffirm the notion that there is still work to be done in terms of the prevention and control of *Leishmania* infection, even with the growing number of new insecticide formulations for CanL prevention over the previous 10 years. Dogs’ vaccination do not prevent *Leishmania* infection but rather the reduction of clinical signs compatible with the disease, which is crucial for the management of CanL. Nevertheless, the animal represents a focus of infection in terms of public health, and as such, all vaccinated dogs, whether infected or not, must use effective insecticide/repellent against sand flies, as it is the only measure that prevents infection and blocks transmission.

In areas with high prevalence levels, increasing the awareness regarding the compliance and proper application of repellents as well as avoiding conditions that enable sand flies’ breeding, such as accumulated trash or organic material in corrals, chicken coops or hutches where dogs also roam, may also contribute to the control of this zoonosis. Not less important are dogs that do not have access to veterinary care, or with limited access, such as sheltered and abandoned dogs. The results of the study only reflect what is happening at the level of dogs whose owners take them to the veterinarian; all the others were left out. These dogs will not have access to vaccines, will not be protected with insecticides, will be undernourished and therefore will be an easy feeding source for the vector. Nevertheless, all veterinarians play a key role in the continuous surveillance of suspected CanL cases and prevention of this zoonosis. Further research and continuous monitoring of leishmaniasis at human and veterinary level need to be done. Epidemiological surveys as well as entomological studies are of the utmost importance and should be promoted periodically in endemic regions.

## Figures and Tables

**Figure 1 microorganisms-10-02262-f001:**
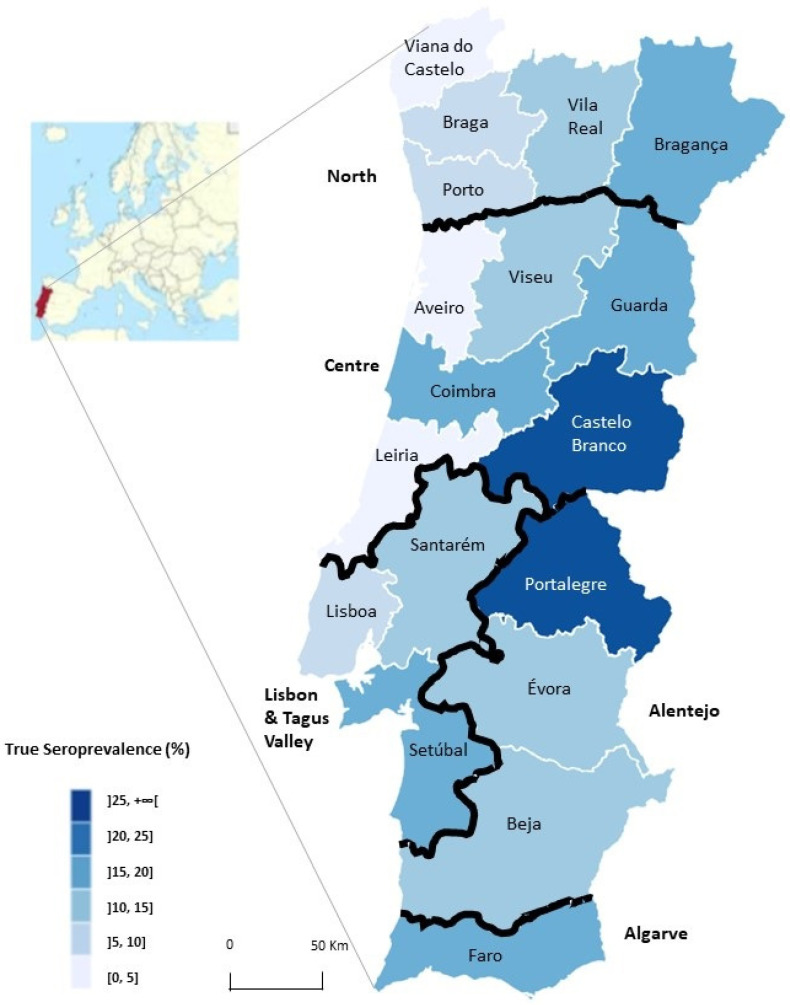
Map of continental Portugal showing a categorical representation of true seroprevalence for *Leishmania* infection in dogs determined by DAT, per district and NUTS2. (Map draw in mapinseconds.com (accessed on 16 October 2022)).

**Figure 2 microorganisms-10-02262-f002:**
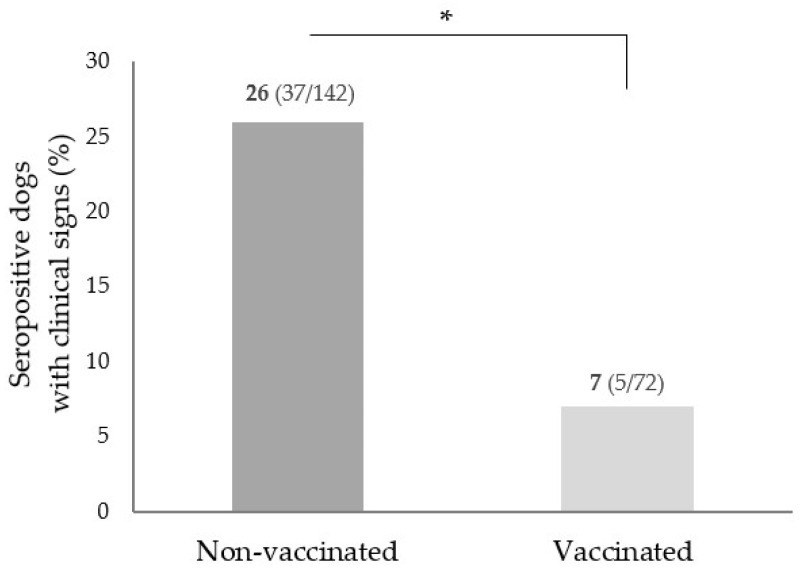
Percentage of seropositive dogs with clinical signs compatible with canine leishmaniosis in both nonvaccinated and vaccinated groups. * *p* = 0.002.

**Table 1 microorganisms-10-02262-t001:** Number of enrolled veterinary clinics, and seroprevalence by region (NUTS2) / district of the whole sample set.

Region/District	No. Veterinary Clinics	No. Dog Samples	No. Seropositive Dogs (%)	% True Seroprevalence	95% CI
**Littoral/Interior**					
Littoral	59	1076	99 (9.2)	9.7	8.0–11.7
Interior	39	784	118 (15.1) *	15.8	13.4–18.7
**North**	26	458	42 (9.2)	9.6	7.2–12.8
Braga ^1^	6	93	6 (6.5)	6.9	3.2–14.4
Bragança ^2^	5	82	12 (14.6)	15.7	9.2–25.9
Porto ^1^	7	128	11 (8.6)	9.2	5.2–15.8
Viana do Castelo ^1^	3	51	0 (0)	0	0.0–7.5
Vila Real ^2^	5	104	13 (12.5)	13.4	8.0–21.7
**Centre**	31	529	63 (11.9)	12.5	9.9–15.7
Aveiro ^1^	5	89	1 (1.1)	1.2	0.0–6.4
Castelo Branco ^2^	3	72	20 (27.8)	29.9	20.1–42.0
Coimbra ^1^	8	131	20 (15.3)	16.4	10.9–24.1
Guarda ^2^	2	40	7 (17.9)	19.3	9.6–35.1
Leira ^1^	7	110	4 (3.6)	3.9	1.5–9.6
Viseu ^2^	6	87	11 (12.8)	13.8	7.8–23.1
**Lisbon and Tagus Valley**	25	496	53 (10.7)	11.2	8.7–14.4
Lisboa ^1^	12	210	18 (8.6)	9.2	5.9–14.1
Santarém ^2^	7	161	17 (10.6)	11.4	7.2–17.5
Setúbal ^1^	6	125	18 (14.4)	15.5	10.0–23.2
**Alentejo**	11	247	37 (15.3)	16.1	11.9–21.4
Beja ^2^	4	60	8 (13.3)	14.3	7.4–26.0
Évora ^2^	5	122	12 (9.8)	10.6	6.2–17.6
Portalegre ^2^	2	60	17 (28.3)	30.5	19.9–43.8
**Algarve**	5	135	22 (16.3)	17.2	11.8–24.7
Faro ^1^	5	135	22 (16.3)	17.2	11.8–25.2
**Total**	98	1860	217 (11.7)	12.5	10.3–13.2

^1^ Littoral district; ^2^ Interior district; * *p* < 0.001; CI, confidence interval.

**Table 2 microorganisms-10-02262-t002:** Number of samples and dogs’ seropositivity per sex, age, breed, fur size, housing, repellents/ insecticide use, vaccination status and presence of canine leishmaniosis clinical signs.

Variables	No. Samples (%)	No. Seropositive Dogs (%)	95% CI
**Sex ^1^**	1834		
Female	944 (51.5)	105 (11.1)	9.1–13.1
Male	890 (48.5)	111 (12.5)	10.3–14.7
**Age group (years) ^2^**	1731		*
(0.5–2)	506 (29.2)	36 (7.1)	0.0–15.4
(3–5)	465 (26.9)	64 (13.8)	5.3–22.2
(6–8)	387 (22.4)	51 (13.2)	3.9–22.5
(9–11)	243 (14.0)	27 (11.1)	0.0–22.9
(12–17)	130 (7.5)	20 (15.4)	0.0–31.2
**Breed** Non-autochthonous pure breed Autochthonous pure breed Crossbreed Mongrel dogs	1860		
	819 (43.9)	104 (12.7)	6.3–19.1
	104 (5.9)	15 (14.4)	0.0–32.2
	165 (8.9) 772 (41.6)	12 (7.3) 86 (11.1)	0.0–22.0 4.4–17.7
**Fur size ^3^**	1841		
Short	1109 (60.2)	135 (12.2)	6.7–17.7
Medium	505 (27.4)	57 (11.3)	3.1–19.5
Long	227 (12.3)	24 (10.6)	0.0–22.9
**Dog’s housing ^4^**	1835		
Exclusively outdoors	227 (12.4)	22 (9.7)	0.0–22.1
Mostly outdoors	637 (34.7)	73 (11.5)	4.2–18.8
Equally indoors and outdoors	323 (17.6)	33 (10.2)	0.0–20.5
Mostly indoors	252 (13.7)	31 (12.3)	0.1–23.9
Exclusively indoors	396 (21.6)	58 (14.6)	5.5–23.7
**Repellents/ insecticides ^5^**	1825		
Effective	732 (40.1)	90 (12.3)	5.5–19.1
Noneffective	295 (16.2)	26 (8.8)	0.0–19.7
Use but unknown	242 (13.3)	25 (10.3)	0.0–22.2
Nonuse	556 (30.4)	73 (13.1)	5.3–20.8
**Vaccination status ^6^**	1824		*
Vaccinated with LetiFend^®^	142 (7.8)	26 (18.3)	3.4–33.2
Vaccinated with CaniLeish^®^	63 (3.5)	30 (47.6)	29.7–65.5
Vaccinated (unknown vaccine)	66 (3.6)	16 (24.2)	3.2–45.2
Nonvaccinated	1553 (85.1)	142 (9.1)	4.3–13.8
**CanL clinical signs ^7^**	1804		*
Yes	112 (6.2)	42 (37.5)	22.8–52.1
No	1692 (93.8)	175 (10.3)	5.8–14.8

^1^ 26 samples with no data on sex; ^2^ 129 samples with no data on age and 0.5 corresponds to dogs with 6 months old; ^3^ 19 samples with no data on fur size; ^4^ 25 samples with no information on living environment; ^5^ 35 samples with no data on use of repellents/insecticides; ^6^ 36 samples with no data on vaccination status; ^7^ 56 samples with no data on existence of CanL compatible clinical signs; * Statistically significant variable (*p =* 0.001); CI, confidence interval.

**Table 3 microorganisms-10-02262-t003:** Potential risk factors for Leishmania infection excluding vaccinated dogs, according to logistic regression models to estimate crude and adjusted odds ratio values.

Risk Factor *	Univariate			Multivariate		
	% in Sample	Crude OR	95% CI	Adjusted OR	95% CI	*p*-Value
Older than 2 years	30.1	1.60	1.0–2.5	1.68	1.1–2.6	0.02
Residing in the Interior	42.5	2.21	1.6–3.1	1.92	1.3–2.9	0.002
Residing in Alentejo	11.5	2.01	1.3–3.1	--	--	--
Nonuse of repellents/insecticides	30.9	1.60	1.1–2.3	1.75	1.2–2.5	0.003
Constant				0.050		<0.001
Hosmer and Lemeshow Test				Sig. = 0.811		

* Only statistically significant variables are presented (*p* ≤ 0.05).

## Data Availability

Not applicable.

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
