# Peer review of "Seroprevalence and Risk Factors Associated with Leishmania Infection in Dogs from Portugal"

_microorganisms, 2022, doi:10.3390/microorganisms10112262_

Round 1
Reviewer 1 Report
The manuscript, Seroprevalence and Risk Factors Associated with Leishmania Infection in Dogs from Portugal, provides a needed epidemiological update on this highly relevant zoonotic infection in Portugal. The authors present a well-structured study that contains the necessary information to understand the rationale, design, and main observations associated with risk assessment and vaccination. The study also addresses the main inherent limitations that contain, providing a relevant source of epidemiological information for the field.
Title:
The title is well constructed and convenes well the message of the manuscript.
Abstract:
The abstract is easy to read and structures well the content of the manuscript. The authors might include in the Abstract “Continental Portugal” and mention the cohort as “client-owned dogs with access to veterinarian care” for more accurate geographical representation and immediate cohort recognition.
Introduction:
The introduction is adequately providing a solid understanding of the overall epidemiological situation, the need for the work, and the main goals. It would profit the reader and improve the accuracy of the report by adding one line about the fact that the study is done on continental Portugal. Azores and Madeira are not included and their epidemiological condition is distinct from the rest of the country.
The Material and Methods:
The Material and Methods section is well-written and presents sufficient detail for the reader.
I would like the authors to comment if domperidone (Leisguard®), associated with reduced antibody formation can contribute to increased seronegativity? How often are dogs given domperidone in Portugal? Do you have any insights if this might be a relevant factor for seropositivity determination? If relevant, please add to the discussion.
Result:
The Results section although clear and well-structured contains minor issues. For example, there is no table 3 in the manuscript, but there is table 4.
In figure 1, I could not understand if the seroprevalence in the map was associated with a colour gradient or if it was a categorical representation. If it is a continuous colour gradient, the higher the seroprevalence the darker in a continuous colour increment. If it is a categorical representation all the samples in the ranges defined should have the same colour. Looking at table 1 have difficulty adjusting the % of true seroprevalence data to the map in figure 1. For example, Faro 17.2% is much darker than Coimbra 16.4% or Setubal 15.7%. (The difference is less than 2%). Santarém with 11.4% is similar in color to Coimbra or Setubal (The difference is more than 4-5%). Lisbon is 9.1% almost the same as Setubal and is clearly less dark. Just taking these examples, neither a categorical nor a continuous gradient explains the colours. Maybe there is a more elaborate statistical explanation for this that I am not seeing. If there is a such explanation, please describe it in the caption. Considering that these maps are often used as a reference, it would be essential clarification to the reasoning for the colouring in the map and describe it in the caption. Also, include in the Caption “seroprevalence for Leishmania infection determined by DAT”.
The authors make a significant effort to address the issue of vaccination. In line 214 the authors comment on the differences in of % vaccinated animals in the country (report data not shown). Considering the quality of the data provided, the reporting of vaccinated animals' geographical distribution would profit the manuscript. The reasons for this are two-fold. First is just the reporting of vaccination coverage. The second is to better understand the relevance of the data reported in table 2 concerning vaccination. One of the significant differences reported in table 2 is the (expected) CaniLeish increased seropositivity. How was the distribution of vaccination by geographical region? If there was a strong regional bias of a specific vaccination approach for regions with higher (or lower) seropositivity? Although this explanation is not likely, disclosing the vaccination distribution would rule out this possibility.
The authors report that Letifend is not associated with cross-reactivity, the % of seropositive animals reported in table 2 is 18.6%, which this higher than the national average, and also double the of non-vaccinated animals (9.1%). Once again reporting the geographic origin of the vaccinated animals would contribute to a better understanding of the data. Can you comment on this?
I understand that the seropositive animals vaccinated with CaniLeish might be due to DAT cross-reactivity with the vaccine, but if the animals are seronegative, why are they excluded? What is the rationale to exclude CaniLeish seronegative animals from Table 4? Irrelevantly of their vaccination status, they are seronegative, correct? Using this rationale to me makes also excludes animals with unknown vaccination status. Looking at the data in table 2, the average seropositivity for non-vaccinated animals is lower than for vaccinated animals. Thus, excluding clinical signs, from table 2 vaccination was associated with the highest seropositivity (even excluding CaniLeish). In table 4 vaccination was evaluated as a Risk Factor for seropositivity. Would the risk assessment in table 4 be more appropriate if excluded all vaccinated animals?
Discussion:
The discussion section is adequate. It contains several relevant comparisons that enable a good integration of the data reported in the available literature. Several studies are mentioned. Considering that several reports have highlighted issues with concordance between different tests, see for example the Scientific reports by Santarem et al in 2020 (Sci Rep. 2020 May 28;10(1):8988. doi: 10.1038/s41598-020-66088-5). Thus, it would profit the reader that when a study is mentioned the serological approach should also be included, for example in line 329 the study is performed using ELISA with promastigote extract. This is relevant information to the reader. In fact, the author's rational for the use of DAT to enable a more accurate comparison with a previous study was perfect and recognizes this variability, thus it makes sense that all comparisons include the reference to the approach, at least in the discussion section. I would also like to discuss the impact of vaccination on serological surveys. Although the number of vaccinated animals was not sufficient to provide statistical power to the comparison between vaccinated with LetiFend (only 142) and non-vaccinated the data provided does not make a solid argument that this vaccination regimen does not have an impact in DAT serological surveys. Once again geographic distribution might help clarify this apparent bias. Please comment on the discussion.
Considering how small the country is, and the absence of information about mobility in the inquiry presented, what is the opinion of the author on the impact of trans-regional mobility in the distribution of seroprevalence. Could you comment this in the discussion?
Minor corrections/recomendations:
Normally the term leishmaniosis is most used to describe the disease in dogs, while leishmaniasis is most often used for the human disease. This is might be found in official WHO or CDC documentation. Still, these terms are often used interchangeably and inconsistently used in the literature. This was not the case for the authors that follow the recommendation of article 53 by G. Miro and use only Leishmaniosis, which for me is acceptable although I prefer to use leishmaniasis for the human form of the disease. There is also an obvious impact that the absence of leishmaniasis in the key words (or in the text) can limit access to the manuscript by people who research “Leishmaniasis”. Thus I would recommend adding leishmaniasis, if not to the text when mentioning the human disease, at least to the keywords. Obviously, this is only a recommendation it has no impact on the quality of the manuscript.
In the keywords there should be consistency, some words are capitalized others are not.
There is a typo in the keywords “Southwester Europe” – Should read Southwestern.
Districts and the associated words like Littoral or Interior are not consistently written, sometimes capitalized other times not. For example, “Interior Districts” in lines 330 and 333.
Centre and Center: They are used in the text with different meanings (Centre for the geographic location, and Center for administrative meaning). Center and centre have the same meaning. Center is the correct spelling in American English, while in British English, centre is correct. Please adjust the text for consistency. See also in Figure 1.
Line 80: Suggest replacing “This year (2022)” with “In 2022”
Line 81: I believe that the most accepted DIVA acronym is “differentiating infected from vaccinated animals”. Thus I would suggest rephrasing the sentence to fit the acronym with the proper wording: “LetiFend® is considered a vaccine that enables differentiating infected from vaccinated animals (DIVA)”
Line 140, 261 (Leishmania not italicized)
Line 56 “100.000” notation is not consistent with the decimal separator used in the manuscript “.” Would recommend using 100 000.
In line 156 the number stile used “25000” is not consistent with the rest of the manuscript (please revise the number notation also in tables).
Line 213: Replace Lisbon, with Lisboa.
Line 368 (Replace Leishmania infantum was already defined in the text, suggest replacing with L. infantum)
Line 392 (nonsensical sentence/typo, “In the seroepidemiological This may be”, please rephrase).
Line 415 (nonsensical sentence/typo, “v as LetiFend® is applied as single dose while”, please rephrase).
Table S3 – There is a formatting issue in the “Dog’s population per Region (A)” column, the numbers are not in the column for Alentejo and Algarve.
Reviewer 2 Report
The manuscript, Seroprevalence and Risk Factors Associated with Leishmania Infection in Dogs from Portugal, provides a needed epidemiological update on this highly relevant zoonotic infection in Portugal. The authors present a well-structured study that contains the necessary information to understand the rationale, design, and main observations associated with risk assessment and vaccination. The study also addresses the main inherent limitations that contain, providing a relevant source of epidemiological information for the field.
Title:
The title is well constructed and convenes well the message of the manuscript.
Abstract:
The abstract is easy to read and structures well the content of the manuscript. The authors might include in the Abstract “Continental Portugal” and mention the cohort as “client-owned dogs with access to veterinarian care” for more accurate geographical representation and immediate cohort recognition.
Introduction:
The introduction is adequately providing a solid understanding of the overall epidemiological situation, the need for the work, and the main goals. It would profit the reader and improve the accuracy of the report by adding one line about the fact that the study is done on continental Portugal. Azores and Madeira are not included and their epidemiological condition is distinct from the rest of the country.
The Material and Methods:
The Material and Methods section is well-written and presents sufficient detail for the reader.
I would like the authors to comment if domperidone (Leisguard®), associated with reduced antibody formation can contribute to increased seronegativity? How often are dogs given domperidone in Portugal? Do you have any insights if this might be a relevant factor for seropositivity determination? If relevant, please add to the discussion.
Result:
The Results section although clear and well-structured contains minor issues. For example, there is no table 3 in the manuscript, but there is table 4.
In figure 1, I could not understand if the seroprevalence in the map was associated with a colour gradient or if it was a categorical representation. If it is a continuous colour gradient, the higher the seroprevalence the darker in a continuous colour increment. If it is a categorical representation all the samples in the ranges defined should have the same colour. Looking at table 1 have difficulty adjusting the % of true seroprevalence data to the map in figure 1. For example, Faro 17.2% is much darker than Coimbra 16.4% or Setubal 15.7%. (The difference is less than 2%). Santarém with 11.4% is similar in color to Coimbra or Setubal (The difference is more than 4-5%). Lisbon is 9.1% almost the same as Setubal and is clearly less dark. Just taking these examples, neither a categorical nor a continuous gradient explains the colours. Maybe there is a more elaborate statistical explanation for this that I am not seeing. If there is a such explanation, please describe it in the caption. Considering that these maps are often used as a reference, it would be essential clarification to the reasoning for the colouring in the map and describe it in the caption. Also, include in the Caption “seroprevalence for Leishmania infection determined by DAT”.
The authors make a significant effort to address the issue of vaccination. In line 214 the authors comment on the differences in of % vaccinated animals in the country (report data not shown). Considering the quality of the data provided, the reporting of vaccinated animals' geographical distribution would profit the manuscript. The reasons for this are two-fold. First is just the reporting of vaccination coverage. The second is to better understand the relevance of the data reported in table 2 concerning vaccination. One of the significant differences reported in table 2 is the (expected) CaniLeish increased seropositivity. How was the distribution of vaccination by geographical region? If there was a strong regional bias of a specific vaccination approach for regions with higher (or lower) seropositivity? Although this explanation is not likely, disclosing the vaccination distribution would rule out this possibility.
The authors report that Letifend is not associated with cross-reactivity, the % of seropositive animals reported in table 2 is 18.6%, which this higher than the national average, and also double the of non-vaccinated animals (9.1%). Once again reporting the geographic origin of the vaccinated animals would contribute to a better understanding of the data. Can you comment on this?
I understand that the seropositive animals vaccinated with CaniLeish might be due to DAT cross-reactivity with the vaccine, but if the animals are seronegative, why are they excluded? What is the rationale to exclude CaniLeish seronegative animals from Table 4? Irrelevantly of their vaccination status, they are seronegative, correct? Using this rationale to me makes also excludes animals with unknown vaccination status. Looking at the data in table 2, the average seropositivity for non-vaccinated animals is lower than for vaccinated animals. Thus, excluding clinical signs, from table 2 vaccination was associated with the highest seropositivity (even excluding CaniLeish). In table 4 vaccination was evaluated as a Risk Factor for seropositivity. Would the risk assessment in table 4 be more appropriate if excluded all vaccinated animals?
Discussion:
The discussion section is adequate. It contains several relevant comparisons that enable a good integration of the data reported in the available literature. Several studies are mentioned. Considering that several reports have highlighted issues with concordance between different tests, see for example the Scientific reports by Santarem et al in 2020 (Sci Rep. 2020 May 28;10(1):8988. doi: 10.1038/s41598-020-66088-5). Thus, it would profit the reader that when a study is mentioned the serological approach should also be included, for example in line 329 the study is performed using ELISA with promastigote extract. This is relevant information to the reader. In fact, the author's rational for the use of DAT to enable a more accurate comparison with a previous study was perfect and recognizes this variability, thus it makes sense that all comparisons include the reference to the approach, at least in the discussion section. I would also like to discuss the impact of vaccination on serological surveys. Although the number of vaccinated animals was not sufficient to provide statistical power to the comparison between vaccinated with LetiFend (only 142) and non-vaccinated the data provided does not make a solid argument that this vaccination regimen does not have an impact in DAT serological surveys. Once again geographic distribution might help clarify this apparent bias. Please comment on the discussion.
Considering how small the country is, and the absence of information about mobility in the inquiry presented, what is the opinion of the author on the impact of trans-regional mobility in the distribution of seroprevalence. Could you comment this in the discussion?
Minor corrections/recomendations:
Normally the term leishmaniosis is most used to describe the disease in dogs, while leishmaniasis is most often used for the human disease. This is might be found in official WHO or CDC documentation. Still, these terms are often used interchangeably and inconsistently used in the literature. This was not the case for the authors that follow the recommendation of article 53 by G. Miro and use only Leishmaniosis, which for me is acceptable although I prefer to use leishmaniasis for the human form of the disease. There is also an obvious impact that the absence of leishmaniasis in the key words (or in the text) can limit access to the manuscript by people who research “Leishmaniasis”. Thus I would recommend adding leishmaniasis, if not to the text when mentioning the human disease, at least to the keywords. Obviously, this is only a recommendation it has no impact on the quality of the manuscript.
In the keywords there should be consistency, some words are capitalized others are not.
There is a typo in the keywords “Southwester Europe” – Should read Southwestern.
Districts and the associated words like Littoral or Interior are not consistently written, sometimes capitalized other times not. For example, “Interior Districts” in lines 330 and 333.
Centre and Center: They are used in the text with different meanings (Centre for the geographic location, and Center for administrative meaning). Center and centre have the same meaning. Center is the correct spelling in American English, while in British English, centre is correct. Please adjust the text for consistency. See also in Figure 1.
Line 80: Suggest replacing “This year (2022)” with “In 2022”
Line 81: I believe that the most accepted DIVA acronym is “differentiating infected from vaccinated animals”. Thus I would suggest rephrasing the sentence to fit the acronym with the proper wording: “LetiFend® is considered a vaccine that enables differentiating infected from vaccinated animals (DIVA)”
Line 140, 261 (Leishmania not italicized)
Line 56 “100.000” notation is not consistent with the decimal separator used in the manuscript “.” Would recommend using 100 000.
In line 156 the number stile used “25000” is not consistent with the rest of the manuscript (please revise the number notation also in tables).
Line 213: Replace Lisbon, with Lisboa.
Line 368 (Replace Leishmania infantum was already defined in the text, suggest replacing with L. infantum)
Line 392 (nonsensical sentence/typo, “In the seroepidemiological This may be”, please rephrase).
Line 415 (nonsensical sentence/typo, “v as LetiFend® is applied as single dose while”, please rephrase).
Table S3 – There is a formatting issue in the “Dog’s population per Region (A)” column, the numbers are not in the column for Alentejo and Algarve.
